# The Shear Stress–Regulated Expression of Glypican-4 in Endothelial Dysfunction In Vitro and Its Clinical Significance in Atherosclerosis

**DOI:** 10.3390/ijms241411595

**Published:** 2023-07-18

**Authors:** Katharina Urschel, Karsten P. Hug, Hanxiao Zuo, Michael Büttner, Roman Furtmair, Constanze Kuehn, Florian M. Stumpfe, Balaz Botos, Stephan Achenbach, Yan Yuan, Barbara Dietel, Miyuki Tauchi

**Affiliations:** 1Department of Medicine 2—Cardiology and Angiology, Universitätsklinikum Erlangen, Friedrich-Alexander-Universität Erlangen-Nürnberg (FAU), Schwabachanlage 12, 91054 Erlangen, Germany; katharina.urschel@uk-erlangen.de (K.U.); hug@dhm.mhn.de (K.P.H.); roman.furtmair@uk-erlangen.de (R.F.); stephan.achenbach@uk-erlangen.de (S.A.); barbara.dietel@uk-erlangen.de (B.D.); 2School of Public Health, University of Alberta, 11405 87 Avenue, Edmonton, AB T6G 1C9, Canada; hzuo3@ualberta.ca (H.Z.); yyuan@ualberta.ca (Y.Y.); 3Department of Obstetrics and Gynaecology, Universitätsklinikum Erlangen, Friedrich-Alexander University Erlangen-Nürnberg (FAU), Universitätsstraße 21-23, 91054 Erlangen, Germany; florian.stumpfe@uk-erlangen.de; 4Department of Vascular and Endovascular Surgery, General Hospital Nuremberg, Paracelsus Medical University, Breslauer Str. 201, 90471 Nuremberg, Germany; balazs.botos@klinikum-nuernberg.de

**Keywords:** glypican-4, heparan sulfate, endothelial dysfunction, atherosclerosis, plaque, shear stress, sex differences

## Abstract

Retention of circulating lipoproteins by their interaction with extracellular matrix molecules has been suggested as an underlying mechanism for atherosclerosis. We investigated the role of glypican-4 (GPC4), a heparan sulfate (HS) proteoglycan, in the development of endothelial dysfunction and plaque progression; Expression of GPC4 and HS was investigated in human umbilical vein/artery endothelial cells (HUVECs/HUAECs) using flow cytometry, qPCR, and immunofluorescent staining. Leukocyte adhesion was determined in HUVECs in bifurcation chamber slides under dynamic flow. The association between the degree of inflammation and GPC4, HS, and syndecan-4 expressions was analyzed in human carotid plaques; GPC4 was expressed in HUVECs/HUAECs. In HUVECs, GPC4 protein expression was higher in laminar than in non-uniform shear stress regions after a 1-day or 10-day flow (*p* < 0.01 each). The HS expression was higher under laminar flow after a 1 day (*p* < 0.001). Monocytic THP-1 cell adhesion to HUVECs was facilitated by GPC4 knock-down (*p* < 0.001) without affecting adhesion molecule expression. GPC4 and HS expression was lower in more-inflamed than in less-inflamed plaque shoulders (*p* < 0.05, each), especially in vulnerable plaque sections; Reduced expression of GPC4 was associated with atherogenic conditions, suggesting the involvement of GPC4 in both early and advanced stages of atherosclerosis.

## 1. Introduction

Atherosclerosis is characterized by the accumulation of lipids and inflammatory cells within the intima of the arterial wall, leading to plaque formation. The retention of atherogenic lipoproteins in the subendothelial matrix through interaction with extracellular matrix molecules, particularly proteoglycans, has been suggested as a key underlying mechanism [1,2]. Under pathological conditions, lipoproteins undergo alterations such as oxidation, glycation, hydrolysis, and sulfation, which promote inflammation and enhance the atherogenic process [3].

Atherosclerotic plaque formation frequently occurs at arterial bifurcations, where the blood flow pattern is turbulent, generating non-uniform shear stress [4]. Turbulent flow patterns at the plaque shoulder (PS) regions evoke an increased inflammatory response, which in turn seems to increase the likelihood of plaque rupture [5,6,7]. Consequently, endothelial cells (ECs) in atheroprone regions express higher levels of inflammatory triggers, which promote plaque vulnerability, leading to plaque rupture or plaque erosion [8].

The luminal surface of vascular ECs is covered by the glycocalyx, an extracellular layer of 1–5 µm thickness, consisting of glycosaminoglycans, proteoglycans, and glycoproteins [9]. The height of adhesion receptors that bind to leukocytes, such as selectins and integrins, ranges from 20 to 40 nm. Hence, they are shorter than the thickness of the glycocalyx [10]. The glycocalyx is a barrier for leukocytes and prevents them from rolling on and adhering to the EC monolayer. Therefore, an intact glycocalyx is considered to be atheroprotective. Furthermore, heparan sulfate (HS), a glycosaminoglycan component of the surface glycocalyx layer, has many functions that may influence atherogenesis. For example, endothelial HS responds to shear stress, which in turn influences nitric monoxide (NO) production of ECs [11]. HS also modulates the signal strength of fibroblast growth factors (FGFs), which can aggravate atherogenesis [12,13]. Furthermore, HS influences the subendothelial retention of lipoproteins by actively binding to lipoproteins, as well as by regulating the binding between lipoproteins and ECs [14,15]. Furthermore, mRNA levels of heparanase, a degradation enzyme of HS, are higher in human carotid arteries bearing plaques than in healthy controls [16], which suggests an involvement in atherogenesis. The major HS proteoglycans found in the glycocalyx of cardiovascular ECs include glypicans, syndecans, and perlecan [17]. Of these, the deletion of syndecan-1 (SDC1) or syndecan-4 (SDC4) has been shown to promote atherosclerotic plaque formation, even in regions that are normally protected by laminar blood flow [18,19].

However, only a limited number of proteoglycans have been investigated to date. Particularly, the role of glypican-4 (GPC4) remains unclear. So far, it has been shown to be expressed in the endothelial lining of blood vessels in synovia, with a stronger expression in an inflammatory state [20]. GPC4 as an adipokine regulates insulin receptor signaling [21] and thus may be associated with cardiovascular disease through obesity, but it is not known whether endothelial GPC4 expression plays a role in atherogenesis. Furthermore, GPC4 is encoded on the X-chromosome (Xq26.2) and can escape X-chromosomal inactivation; therefore, sex-dependent differences in GPC4 expression might exist, as seen in other genes of this group [22,23]. A relevant role of GPC4 in atherosclerosis could therefore help in understanding the observed gender differences in the prevalence of cardiovascular diseases.

We hypothesized that endothelial GPC4 expression plays a role in atherogenesis by modulating endothelial activation and immune cell adhesion. 

## 2. Results

### 2.1. HUVECs and HUAECs Expressed GPC4 

We first confirmed that GPC4 is expressed in human umbilical artery ECs (HUAECs) and human umbilical vein ECs (HUVECs) using flow cytometry. Cells were gated on a forward scatter (FSC)-A and side scatter (SSC)-A plot (Figure 1A). Cells in the gate (mean ± standard deviation) were 31.27 ± 9.93% in HUAECs and 42.64 ± 10.16% in HUVECs. The analyzed cells were confirmed to express CD31 (94.88 ± 8.48% in HUAECs and 97.70 ± 2.29% in HUVECs) and to be propidium iodide-negative. The expression of GPC4 was detected by antibody staining in both HUAECs and HUVECs (see representative histogram in Figure 1B, dark green). 

Figure 1C shows mean fluorescence intensity of measured cells, which were stained with 12 antibodies (Appendix A; female *n* = 6 and male *n* = 7). CDs 31 and 146 served as positive, and CD45 as negative controls. Two samples were excluded from further analysis due to unstained positive controls (Figure 1C, marked with “X”). CD62P, CD62E, CD106, CD54, and CD105 were detected, although ECs were not stimulated for activation. There was no consistent trend in expression difference between days 0 and 7 in either EC types. Mean fluorescence intensity of these activation markers in HUVECs were higher than those in HUAECs (*p*-values for CD62P, *p* < 0.001 on day 0; CD106, *p* = 0.033 on day 7; CD54, *p* < 0.001 on day 0 and *p* = 0.044 on day 7; and CD105, *p* < 0.001 on day 0 and day 7) (data for CD106, CD54, and CD105 were log-transformed for statistical purposes). The correlations between the HUVEC marker CD31 and these activation markers were statistically significant in HUVECs (CD62P, *p* = 0.002; CD62E, *p* = 0.02; CD54, *p* = 0.01 on day 0), while a trend was shown in HUAECs (CD62P, *p* = 0.642; CD62E, *p* = 0.474; CD54, *p* = 0.266 on day 0). GPC4 expression, regardless of the incubation time or sex, was higher in HUVECs than in HUAECs (*p* = 0.024; Figure 1D), which is attributed to the difference on day 0 (*p* = 0.001; Figure 1E).

The expression levels of mRNA were investigated in HUAECs and HUVECs (*n* = 9 each) incubated for either 0 or 7 days after confluence was reached, either with or without tumor necrosis factor (TNF)-α stimulation (Figure 2). Most of the analyzed genes (26 genes in total) showed within five-fold differences between incubation times (day 7 vs. day 0) and stimulation conditions (treated with vs. without TNF-α) in either HUAECs or HUVECs (Figure 2A). Figure 2 shows the fold-increase (red) or -decrease (green) of gene expressions on day 7 relative to those on day 0 (Figure 2B) and after TNF-α stimulation relative to those without TNF-α stimulation (Figure 2C). The genes that belong to the functional group of HS-/CS-proteoglycans showed a trend for higher expression on day 7, as did the degradation enzyme MMP2 (Figure 2B, Appendix A). Most investigated genes were not responsive to TNF-α stimulation. Within the HS-/CS-proteoglycans functional group, only SDC4 expression (framed in yellow line, Figure 2C) increased by TNF-α stimulation consistently in all samples, regardless of the incubation time. GPC4 expression (framed in pink line in Figure 2B,C) did not increase by TNF-α stimulation (Figure 2C) but by 7-day incubation (Figure 2B), especially in HUVECs. Analyses suggested that GPC4 expression may be correlated with differential expression of cholesterol transport-related genes both in days 7 vs. 0 and in conditions with vs. without TNF-α stimulation (Appendix A).

### 2.2. GPC4 Expression Was Regulated by Shear Stress 

HUVECs were exposed to flow for either 1 day or 10 days, and GPC4 and HS were visualized using immunocytochemical staining (Figure 3A). In preliminary experiments, expression of GPC4 and HS was not different between HUAECs and HUVECs: therefore, the dynamic flow experiments were performed using HUVECs only. We observed a positive correlation between fluorescence intensities of immune labeled GPC4 and HS at laminar (*p* = 0.003; *r* = 0.770) and non-uniform (*p* = 0.002, *r* = 0.803) regions after a 1-day flow (*n* = 12), which was weakened after 10 days (laminar: *p* = 0.01; *r* = 0.689, non-uniform: *p* = 0.1, *r* = 0.497).

GPC4 showed significantly lower expression at non-uniform compared with laminar flow areas after both 1-day and 10-day flows (1 day, *p* = 0.008, *n* = 24; 10 days, *p* = 0.003, *n* = 25; Figure 3B, left panel). There was also an overall tendency of decreasing GPC4 expression with longer flow (*p* = 0.12). The corrected total cell fluorescence (CTCF, see Section 4.4) of HS was significantly different between regions of laminar and non-uniform shear stress after the 1-day flow (*p* < 0.001), but the difference was absent after the 10-day flow (Figure 3B, right panel). Furthermore, we observed flow-dependent differences in the HS expression pattern. After the 1-day flow, we observed a reticular pattern in laminar flow regions that was continuously disrupted in regions of non-uniform shear stress. After 10-day incubation, the pattern was disrupted in both regions (Figure 3C).

### 2.3. GPC4 Knockdown in HUVECs Facilitated THP-1 Cell Adhesion

After 1-day or 10-day flow exposure, HUVECs were stimulated with TNF-α, and adhesion of THP-1 cells was quantified (*n* = 29 for 1-day and *n* = 28 for 10-day exposure). The number of adhered THP-1 cells was significantly lower in regions of laminar compared with non-uniform shear stress in both exposure periods (*p* < 0.001 for each, Figure 4A). The number of adhered THP-1 cells in non-uniform regions tended to be higher after the 10-day flow than after the 1-day flow (*p* = 0.16, Figure 4A). 

To investigate whether endothelial GPC4 plays a role in THP-1 cell adhesion, GPC4 was knocked down in HUVECs by siRNA transfection, and adhesion assays were performed. By transfecting siRNA, GPC4 expression decreased to approximately 60% (Appendix A). THP-1 cell adhesion was significantly higher in HUVECs transfected with GPC4 siRNA than in those transfected with scramble siRNA in regions of laminar (*p* = 0.03) and non-uniform shear stress (Figure 4B, *p* < 0.001), while the expression of VCAM-1 and E-selectin was not different (Figure 4C and D, respectively). Treatment with the HS degradation enzyme heparinase III reduced HS in HUVECs (Appendix A) but showed no effect on THP-1 monocyte adhesion (Figure 4E). Furthermore, heparinase III treatment on *GPC4* siRNA-transfected ECs had no additional effect on THP-1 cell adhesion. Additionally, the HS antagonist surfen hydrate showed no effect on THP-1 cell adhesion under flow conditions (Figure 4F).

### 2.4. Sex Differences in GPC4

We further analyzed possible sex differences in GPC4 expression in our in vitro setting. While no sex differences were observed in FACS analyses in ECs cultured under static conditions, we found that GPC4 protein expression tended to be higher in female compared to male samples under flow conditions in laminar and non-uniform shear stress regions (Figure 4G; female *n* = 11, and male *n* = 13). Furthermore, the differences between laminar and non-uniform shear stress regions were more pronounced in females than in males. However, significant differences between males and females were not found for the expression of HS or the adhesion of THP-1 cells. 

### 2.5. Human Atherosclerotic Plaques Lesions—General Characteristics

Patient characteristics are summarized in Table 1. All characteristics analyzed were equally distributed between men and women. There were no significant differences between men and women in any characteristics analyzed. Included patients had an average stenosis of 83.15 ± 1.0%. The degree of stenosis was higher in vulnerable compared with stable (*p* < 0.0001) or initial plaque sections (*p* < 0.0001). The size of the NC was the largest in vulnerable plaque sections and was significantly smaller in stable (*p* = 0.002 vs. vulnerable) or in initial plaque sections (*p* < 0.0001 vs. vulnerable; Figure 5A,B). Furthermore, the FC was significantly thicker in stable compared with vulnerable plaque sections (*p* = 0.003; Figure 5C). The degree of stenosis significantly correlated with the number of CD68-positive cells at both the more-inflamed PS (*p* = 0.006; *r* = 0.367) and less-inflamed PS (*p* = 0.024; *r* = 0.304, Figure 6). 

### 2.6. Distribution of Glycocalyx Components in Human Atherosclerotic Lesions

The percentage expression of GPC4 (*p* = 0.012), HS (*p* = 0.047), and SDC4 (*p* = 0.059; Figure 7) was lower in the more-inflamed compared with the less-inflamed PS. Correlation analyses showed that the SDC4 and HS expression were positively correlated (Appendix A) both in the more-inflamed PS (*r* = 0.455, *p* < 0.001) and less-inflamed PS (*r* = 0.438, *p* = 0.001). Trends for these positive correlations were also observed between GPC4 and HS (Appendix A) (*r* = 0.241, *p* = 0.08) and for GPC4 and SDC4 (*r* = 0.263, *p* = 0.052; Appendix A) in the less-inflamed PS. Results from the FC regions were comparable to the PS regions. We observed significant positive correlations between the expression of HS and GPC4 (*r* = 0.330, *p* = 0.014) and HS and SDC4 (*r* = 0.277, *p* = 0.04; Appendix A).

### 2.7. Endothelial Expression of Glycocalyx Components Depends on Plaque Vulnerability and Local Inflammation

All plaque sections (*n* = 55) were classified into vulnerable (*n* = 26), stable (*n* = 18), or initial plaque (*n* = 11) sections. The initial plaque sections showed no significant differences in any of the analyzed glycocalyx molecules between both PSs, as demonstrated in Figure 8A. In stable plaque sections, significant differences between the more-inflamed and less-inflamed PS regions were observed in SDC4 (*p* = 0.009) and HS expression (*p* = 0.01), while statistical significance was not reached in GPC4 (*p* = 0.08, Figure 8B). In vulnerable plaque sections, GPC4 was significantly decreased in the more-inflamed PS compared with less-inflamed PS (*p* = 0.03), while HS was less affected (*p* = 0.07). SDC4 showed no significant differences (*p* = 0.3; Figure 8C). 

Finally, the expression pattern of the glycocalyx molecules in the more-inflamed PS of different plaque types were compared. GPC4 and HS showed highest expression in the initial plaque sections compared with vulnerable and stable plaque sections (Figure 9A,B), while SDC4 was not differentially expressed among these plaque types.

Interestingly, strong negative correlations of the glycocalyx molecules and local inflammation determined by CD68 expression in the more-inflamed PS was observed only in vulnerable plaque sections (CD68-SDC4: *r* = −0.416, *p* = 0.030; CD68-GPC4: *r* = −0.356, *p* = 0.07; CD68-HS: *r* = −0.383, *p* = 0.054) but not in initial or stable plaque sections (Appendix A).

## 3. Discussion

The aim of our study was to explore a possible contribution of glycocalyx molecules, especially GPC4 and the associated glycosaminoglycan HS, to atherogenesis. In ECs, expression of various HS-proteoglycans is known, including SDC1, 2, 4, and GPC1, 3, 4 [17,24], but their role in atherosclerosis has not been adequately described thus far. GPC4, SDC4, and HS were expressed in HUVECs and HUAECs under different cell culture conditions. HUVECs showed higher GPC4 levels at day 0 compared to day 7. This expression was regulated by flow: GPC4 expression was higher under atheroprotective laminar flow with high shear stress rates compared with atheroprone non-uniform shear stress, which reflects physiologically atheroprone regions such as bifurcations in vivo. The association between GPC4 and cell types (HUAECs vs. HUVECs), incubation time (day 0 vs. day 7), or cell type–incubation time interaction was not statistically evident. A siRNA-mediated knockdown of GPC4 resulted in higher THP-1 cell adhesion without affecting adhesion molecule expression, while removal of HS had no effect on leukocyte adhesion under flow conditions. Analyses from human atherosclerotic plaque sections revealed a lowered expression of GPC4 and HS in the more-inflamed PS compared with the less-inflamed PS; this was prominent in vulnerable and stable plaque sections and absent in initial lesions. In the correlation analyses for gene expression, the possible correlation between GPC4 and molecules responsible for cholesterol transport was indicated.

It is known that TNF-α stimulation activates HUVECs [25]. In this study, this did not influence the mRNA levels of GPC4 in either HUVECs or HUAECs. The data indicated that GPC4 does not immediately respond to an acute inflammatory stimulus. In the functional group of HS-/CS-proteoglycans, only SDC4 consistently responded to the TNF-α stimulus. While GPC4 expression was not regulated by TNF-α stimulation, it was regulated by shear stress and showed higher expression in the atheroprotective laminar shear stress regions compared with the atheroprone, non-uniform shear stress regions in vitro. These results are in line with those from our human plaque analyses: GPC4 expression in the more-inflamed PS was lower in stable and vulnerable plaque sections than in initial plaque sections. The stable or vulnerable plaques differ from initial plaques in stenosis size; the higher the degree of stenosis, the more turbulent the shear stress pattern, with lowered or oscillatory shear stress, at the PS regions [5,7]. It has been shown that the thickness of the endothelial glycocalyx is shear stress–dependent; for example, higher shear stress rates increased endothelial glycocalyx thickness [26], while low shear stress led to reduced endothelial glycocalyx thickness in mice [27]. Together with these results, our results indicate that GPC4 expression is shear stress–dependent.

In the current study, silencing of GPC4 facilitated THP-1 cell adhesion to TNF-α–activated HUVECs, demonstrating that GPC4 modulates monocyte adhesion and the presence of GPC4 seems to be atheroprotective. Similar results were shown in a previous study, in which GPC1 was studied [28]. Glypicans carry HS glycosaminoglycan, which modulates various biophysiological functions. However, HS in different proteoglycans may have different functions. For example, SDC1, 2, and 4 as well as GPC1 carry HS glycosaminoglycan with a co-receptor function for FGF to strengthen substrate-receptor affinity, but apparently HS in GPC4 does not have this function [29]. Other studies showed that response to tensile forces regarding nitric oxide production by endothelial nitric oxide synthase activation, known to be an anti-inflammatory and antiatherogenic pathway, was mediated by GPC1 and HS, but not by SDC1 or hyaluronic acid [30,31]. In our study, even though GPC4 silencing increased THP-1 cell adhesion, HS degradation using bacterial heparinase III or HS competitive inhibition by surfen hydrate did not facilitate adhesion. Thus, although these proteoglycans have HS in common, different proteoglycans of the endothelial glycocalyx have a distinct role in mechanosignaling.

Among proteoglycans expressed in ECs, SDC4 has been shown to be atheroprotective, and its deletion caused excessive plaque formation in mice [19]. In an in vivo study, combined treatment with heparinase and chondroitinase ABC, but not a single treatment with either, facilitated leukocyte adhesion in HUVECs under flow conditions and TNF-α treatment [32], indicating that CS and HS glycosaminoglycans may have a collaborative role in adhesion regulation. Blocking the extracellular binding capacity of HS by heparin or surfen also failed to affect NO production in ECs, further supporting this hypothesis [33]. Expression of adhesion molecules (i.e., VCAM-1 and E-selectin) was up-regulated by shear stress, in combination with inflammatory stimuli [34]. For this reason, we addressed the question of whether GPC4-knockdown suppresses VCAM-1 and E-selectin expression, thereby decreasing THP-1 cell adhesion. Our results showed that knockdown of *GPC4* did not influence the expression of these adhesion molecules. Similarly, enzymatic degradation of mouse pulmonary endothelial glycocalyx by Heparinase III increased leukocyte adhesion without upregulating adhesion molecules [35]. Furthermore, treatment of activated ECs from human coronary artery with heparinase or chondroitinase ABC showed no effect on adhesion molecule expression [36]. Thus, our results are consistent with these previous studies showing that the endothelial glycocalyx attenuates cell adhesion as a physical barrier without changing the expression of adhesion molecules. On the other hand, previous studies demonstrated that degradation of HS by heparinase-I can inhibit TNF-α–induced monocyte adhesion to the endothelium [37]. Surfen hydrate was also demonstrated to inhibit HS-mediated adhesion of Chinese hamster ovary cells to fibronectin [38], both of which are indicative of the role of HS in adhesion. The reason why HS degradation did not influence cell adhesion in our study is not clear; it may be due to the different cleavage sites by heparinase I vs. III, or different cells used for the experiments. 

The expression of heparanase is increased in human carotid plaques [9]. Our study showed that in human plaque sections, HS and GPC4 expression was lowered in the more-inflamed PS, further indicating an either inflammatory or shear stress–dependent HS degradation in plaque progression. This is in line with data from in vivo studies, where the glycocalyx at atheroprone regions of blood vessels was thinner and associated with plaque initiation [39,40], possibly increasing leukocyte adhesion to ECs [41]. Our findings confirm former studies, as we observed a decrease of glycocalyx molecules in regions of increased inflammation that was more pronounced in lesions with a hemodynamic-relevant stenosis.

Atherosclerosis leads to cardiovascular diseases with severe complications, such as myocardial infraction. Several studies took a closer look at the direct clinical consequences of glycocalyx shedding in patients with cardiovascular disease. Recently, it was shown that increased serum GPC4 levels were highly and significantly associated with an increased risk of major cardiovascular events, vascular mortality, and all-cause mortality, supporting the role of GPC4 shedding in coronary artery disease [42,43]. Furthermore, acute cardiovascular events increased serum SDC1 [44,45], further supporting the hypothesis of shedding as responsible for a reduced glycocalyx expression in coronary artery disease, mainly in acute disorders or advanced atherosclerosis. Recent studies measuring sublingual glycocalyx thickness in different patient cohorts also suggest an association between glycocalyx shedding and atherosclerosis [46] or impaired coronary and myocardial function [47]. Although our results from in vitro cell experiments or ex vivo plaque analyses are not directly comparable, they support the findings from clinical trials and encourage the clinical significance of the glycocalyx in cardiovascular disease.

GPC4 is coded on the X-chromosome (Xq26.2). In a human fibroblast-based assay, two out of nine female samples expressed GPC4 from both X chromosomes, demonstrating that GPC4 could escape X-chromosomal inactivation and have a greater expression in females than in males [23]. In our experimental setting, sex-related differences were not seen in either protein expression by flow cytometry or gene expression by qPCR analysis under static conditions, while higher GPC4 expression was observed in female compared with male ECs under flow conditions. In human atherosclerotic lesions, we did not observe significant differences in GPC4 expression between males and females. The lack of obvious sex differences in GPC4, both in expression and in function in our study, may be explained by the fact that HUVECs are not influenced by circulating sex hormones, which play a central role in sex differences in cardiovascular disease [48]. Furthermore, we analyzed only one section per plaque, which could limit the interpretation regarding sex-specific expression. Additionally, the number of samples, especially female samples, might be too few to detect any differences. 

Among the proteoglycans of the glycocalyx, an atheroprotective role has been demonstrated for GPC1 [28], SDC1 [18], SDC4 [19], biglycan [49], and perlecan [50] in vivo. In contrast, proteoglycans can also be proatherogenic. In the early stages of plaque development in mice, perlecan and biglycan were accumulated with apolipoproteins in the aorta, which may facilitate plaque development by retaining cholesterol in the lesion [50]. Similarly, colocalization of lipids, macrophages, and proteoglycans (biglycan and decorin) were observed in human plaques of an early stage [51]. Furthermore, overexpression of biglycan in mice increased atherosclerosis [52]. These observations support the idea that proteoglycans can contribute to plaque progression by lipid retention. Given the diversity in functions of proteoglycans and glycosaminoglycans, it is difficult to conclude from our results that GPC4 is possibly atheroprotective; however, GPC4 plays a role in leukocyte adhesion in vitro and its degradation during plaque progression. Further research is needed to elucidate the mechanisms as to how and whether GPC4 plays a significant role in atherogenesis.

## 4. Materials and Methods

### 4.1. Cell Culture and Expression Analyses

ECs were freshly isolated from umbilical veins (HUVECs) and arteries (HUAECs) by standard enzymatic digestion techniques [1]. Isolated ECs were cultured in Endothelial Cell Growth Medium (ECGM) with Growth Medium Supplement Mix (PromoCell, Heidelberg, Germany) in a humidified incubator with 7.5% CO_2_ and at 37 °C. 

To analyze expression of GPC4 in ECs by flow cytometry, HUVECs and HUAECs were isolated from the same umbilical cord (female *n* = 6 and male *n* = 7). ECs were seeded in 6-well plates (Falcon, Kaiserslautern, Germany; 4 wells for each cord) with the above-mentioned conditions until a confluence of 80% was reached (day 0); then, 2 wells were further cultured in ECGM without Supplement Mix but with 2% fetal calf serum (FCS; Biochrom, Berlin, Germany) for 7 days to allow ECs to mature (day 7). ECs at day 0 and day 7 were harvested with ice-cold ethylenediaminetetraacetic acid (EDTA) buffer (10 mM in phosphate buffered saline (PBS; Sigma Aldrich, Taufkirchen, Germany)), and a single-cell suspension was prepared at a density of 100,000 cells/100 µL in PBS containing 10% FCS and 2.5% bovine serum albumin (FC buffer). Primary antibodies (Appendix A) and/or isotype controls were added to a final concentration of 1:100 and incubated for 30 min on ice. After washing twice with ice-cold FC buffer, cells were incubated, if necessary, with conjugated secondary antibodies (1:100) for 30 min on ice (staining for GPC4, biglycan, HS, and chondroitin sulfate (CS)). Detailed antibody information is summarized in Appendix A. After washing twice, cells were centrifuged, and the pellets were re-suspended in 250 µL FC buffer for analysis. Flow cytometry was performed using FACSVerse^TM^ (Becton Dickinson, Heidelberg, Germany). Mean fluorescence intensity was calculated from 10,000 cells in total and gated on an FSC-A and SSC-A plot to eliminate debris or dead cells. Mean fluorescent intensity was corrected by subtracting the mean value of the isotype control for each antibody. 

For gene expression analyses, HUVECs and HUAECs (*n* = 9 each) were seeded in 6-well plates and prepared for 0-day and 7-day cultures as described above for flow cytometry analyses. Both 0-day and 7-day cultures were divided into two groups, one of which received 2 h of TNF-α (2.5 ng/mL; Miltenyi Biotec, Bergisch Gladbach, Germany) incubation. Cells were detached with HyQTase™ (Fisher Scientific, Schwerte, Germany). Total RNA was isolated from HUVECs and HUAECs using RNeasy^®^ mini kit (Qiagen, Hilden, Germany) following the manufacturer’s protocol. Primers for 26 genes in 7 functional categories were designed using Primer3Plus (https://www.primer3plus.com/; accessed on 1 June 2023) and custom synthesized (Eurofins Genomics, Ebersberg, Germany). Primer information is summarized in Appendix A. YHWAZ was used as housekeeping control [53]. Real-time quantitative polymerase chain reaction (RT-qPCR) was performed using QuantiTect^®^ Reverse Transcription Kit, QuantiFast^®^ SYBR^®^ Green PCR Kit (Qiagen), and CFX Real-Time System (Bio-Rad Laboratories, Inc., Hercules, CA, USA) with 95 °C denaturation, 60 °C annealing, and 72 °C extension temperatures for 40 cycles. Relative expression levels were calculated using the ΔΔCt method. 

### 4.2. Dynamic Flow Assay

HUVECs were seeded in ibidi^®^ y-shaped µ-Slides (ibidi, Planegg, Germany) with a concentration of 0.5 × 10^6^ cells/mL. After reaching 90–100% confluence, the slides were perfused with ECGM at a flow rate of 9.6 mL/min to achieve a shear stress of 10.2–10.8 dyne/cm^2^ in straight-channel regions (i.e., high shear stress, laminar flow pattern) and 2.5–3.5 dyne/cm^2^ in bifurcation-channel regions (i.e., low shear stress, non-uniform flow pattern) [34]. The laminar flow pattern models physiological atheroprotective hemodynamics, while non-uniform flow pattern with low shear rates models atheroprone regions in the vasculature. Flow was applied for 1 or 10 days.

### 4.3. Dynamic Adhesion Assays

Endothelial activation and immune cell adhesion were investigated using cells exposed to flow. THP-1 monocytic cells (ATCC-TIB-202, Wesel, Germany) were cultured in RPMI 1640 Medium (very low endotoxin, Merck, Darmstadt, Germany) with 1% L-glutamine (Merck, Darmstadt, Germany), 1% penicillin/streptomycin (Gibco, Eggenstein, Germany), and 10% FCS in a humidified incubator with 5% CO_2_ at 37 °C. Flow was applied as described; 3 h prior to the end of flow exposure, ECs were stimulated by adding TNF-α to the media (2.5 ng/mL final concentration) for 2 h, followed by an addition of 7.5 million THP-1 monocytic cells per slide with fresh ECGM for 1 h (Figure 10A,B). 

Dynamic adhesion assay was also carried out using cells in which GPC4 or HS was inhibited. GPC4 was silenced using small interfering (si) RNA. The small interference RNA (siRNA) for GPC4 (Qiagen, catalog number SI03100825) and scrambled siRNAs (Qiagen, catalog number 1027280) were transfected to HUVECs at 60% confluence using Hiperfect^®^ (Qiagen, Hilden, Germany) following the manufacturer’s protocol. Transfection with scrambled siRNA or without siRNA was carried out as negative controls. HUVECs were incubated for 48 h followed by a 1-day flow and adhesion assay (Figure 10C). Bacterial heparinase III (200 mU; Sigma-Aldrich, Taufkirchen, Germany) was added for 4 h to the media to degrade HS, followed by TNF-α stimulation and perfusion with THP-1 monocytes (Figure 10D). Surfen hydrate (5 or 10 µM in dimethyl sulfoxide (DMSO); Sigma-Aldrich, Taufkirchen, Germany) was added as a competitive antagonist of HS for 2 h simultaneously with TNF-α (Figure 10E). As a vehicle control, ECGM only (negative control 1) and 0.02% DMSO in ECGM (negative control 2) were used.

After flow exposure, non-adhering monocytes were rinsed with PBS, and cells were fixed with 4% buffered formalin (Roth, Karlsruhe, Germany) for 10 min at room temperature. Cells were treated with 0.2% Triton X-100 (Sigma-Aldrich, Taufkirchen Germany), stained with hematoxylin (Agilent Waldbronn, Germany,) and eosin (Merk, Darmstadt, Germany), and mounted with Aquatex^®^ (Merck, Darmstadt, Germany). Microphotographs were taken from 6 visual fields of laminar and 8 visual fields of non-uniform shear stress under a light microscope (Olympus, Hamburg, Germany) at 200× magnification with NIS Elements^®^ software version 5.21.03 (Nikon, Düsseldorf, Germany) (Figure 10F). Adhered THP-1 cells were counted in every visual field.

### 4.4. Immunofluorescent Staining

Expression levels of GPC4, HS, vascular cell adhesion molecule (VCAM)-1, and E-selectin were determined using immunofluorescence staining. Detailed information about the antibodies used is summarized in Appendix A. After flow exposure, cells were fixed with ice-cold methanol (Merck, Darmstadt, Germany) for 30 min at −20 °C (GPC4 and HS) or with 4% buffered formalin for 10 min at room temperature (VCAM-1 and E-selectin) and blocked with 5% horse serum in PBS at 37 °C. Cells were incubated with primary antibodies diluted in blocking solution for 1 h at room temperature. Positive staining was visualized by incubating the cells with fluorochrome-conjugated secondary antibodies for 45 min at room temperature. Between procedures, cells were washed well with PBS. Finally, slides were mounted with Fluoroshield^TM^ containing 4′,6-diamidino-2-phenylindole (Sigma-Aldrich, Taufkirchen, Germany).

Microphotographs were taken from regions with laminar and non-uniform shear stress (Figure 10F). Fluorescence intensity was determined using ImageJ^®^ (National Institutes of health, Bethesda, MD, USA) by CTCF as follows: CTCF = integrated density—(area of selected cell × mean fluorescence of background) [54]. 

### 4.5. Patients and Arterial Specimen Collection

Carotid plaque specimens were collected from 55 patients (69% male) undergoing carotid endarterectomy (*n* = 48) or another surgical intervention (*n* = 7). Patient characteristics were pseudo-anonymized and descriptively summarized. The removed tissue samples contained the entire intima and a part of the media of the vessel wall. Plaques were fixed in Bouin’s solution (Sigma-Aldrich, Taufkirchen, Germany), transferred to 70% ethanol (Merk, Taufkirchen, Germany) for at least one week, and subsequently decalcified in EDTA buffer (pH 7.4) for 3 weeks. Afterwards, plaques were embedded in paraffin (Roth, Karlsruhe, Germany) and prepared for immunohistochemical analyses. 

### 4.6. Classification into Initial, Vulnerable, or StablePlaque Sections

All plaque sections were stained with a modified Trichrome staining by Crossman to classify them into vulnerable, stable, or initial plaque sections (Appendix A). Briefly, plaque sections were deparaffinized and hydrated, stained for 10 min (min) with hematoxylin, and subsequently flushed under running tap water. This was followed by acid fuchsine/OrangeG (Merck, Darmstadt, Germany) incubation for 30–60 s and differentiation under microscopic control in 1% phosphomolybdic acid (Merck, Darmstadt, Germany). After washing in distilled water, sections were stained for 5 min in light green (Merck, Darmstadt, Germany), washed again in distilled water, and differentiated in 1% acetic acid (Roth, Karlsruhe, Germany). Sections were dehydrated and mounted with ROTI^®^ Histokitt (Roth, Karlsruhe, Germany). Images were taken at 20× magnification with an Olympus IX70 inverted microscope and analyzed with NIS elements^®^ software version 5.21.03.

Plaque sections were classified based on several parameters, including the size of the necrotic core (NC), neovascularization, plaque rupture and hemorrhage, thickness of the fibrous cap (FC), and infiltrated cells in the PS regions. A plaque with a small necrotic core, leading to less than 55% stenosis, was classified into the “initial” plaque section. Plaques with a degree of stenosis more than 55% were classified into either “stable” or “vulnerable” plaque sections by considering further parameters: plaque sections with a clear sign of rupture were classified into “vulnerable”; those with the thinnest part of the FC > 65 µm were classified into “stable”; those with clear signs of intraplaque hemorrhage in the PS were considered as “vulnerable” [44,55]. Plaque sections were excluded from further analyses if: the plaque sections were not clearly classifiable; >20% of the endothelial lining of the PS was not intact due to erosion or accumulation of erythrocytes; and one or both PSs were damaged for any reason. The classification was performed by a single researcher, and results were blinded for further analyses.

### 4.7. Classification into More- or Less-Inflamed PSs

Each atherosclerotic plaque has an upstream and downstream PS. The upstream PS is more susceptible to inflammation and shows more macrophage infiltration than the downstream PS due to local hemodynamic flow patterns [5]. To classify the PS into more- or less-inflamed PSs, plaques were stained immunohistochemically using anti-CD68 antibody (see Section 4.8). CD68-positive macrophages were counted at a predefined 0.25 mm^2^ region of interest (Appendix A, black rectangles). PSs with more infiltrated CD68-positive cells were defined as the more-inflamed PS, and the other PS as the less-inflamed PS. In the case that the difference was less than 7%, we excluded the plaque sections from further analyses. 

### 4.8. Immunohistochemical Staining

Paraffin-embedded cross-sectional tissues of 4 µm thickness were placed on silane-coated slides (Menzel, Braunschweig, Germany). For the different antibodies, individual staining protocols were established. Two different staining kits were used, with adaptations: CSA (Catalyzed Signal Amplification) II biotin-free Kit for HS and SDC4 antibodies, and CSA Kit (both Agilent, Waldbronn, Germany) for CD68 and GPC4 antibodies. After deparaffinization with ROTI^®^ Histokitt (Roth, Karlsruhe, Germany) and rehydration in isopropanol (Merck, Darmstadt, Germany) to Tris (Tris(hydroxymethyl)aminomethane)-Buffer (Roth, Karlsruhe, Germany), samples were damasked by heating in citrate-buffer (pH 6.0) for 15 min (CD68, HS, SDC4). After cooling for 30 min, plaques were treated with 3% peroxidase block for at least 5 min followed by 30–45 min incubation with a protein block solution. Hence, tissues were incubated at 4 °C overnight with the primary antibody or negative control, with the exception of CD68, which was incubated at room temperature for 15 min. The secondary antibody was incubated at room temperature for 15–30 min, depending on the primary antibody (Appendix A). Using the CSA biotinylated kit, slides were then incubated with a streptavidin–biotin complex for 30 min, followed by an amplification step and treatment with streptavidin–peroxidase for 15 min. Visualization was done using 3,3-diaminobenzidine (DAB) under microscopic control. Using the CSA II biotin-free Kit, plaques were incubated after the secondary antibody for 15 min with an amplification reagent, followed by anti-fluorescein coupled with horseradish peroxidase. Visualization of positively stained tissue was done using liquid DAB under microscopic control. With the exception of before the primary antibody, slides were carefully washed before each step with Tris-buffer (pH 7.6) with 0.1% Tween 20 (Merck, Darmstadt, Germany), as recommended in the manufacturer’s protocol. All sections were counterstained with hematoxylin for 5–10 min and mounted with Aquatex^®^ (Merck, Darmstadt, Germany). 

Pictures were taken with an inverted light microscope (Olympus IX70) at 150× magnification with NIS elements^®^ software version 5.21.03. From each plaque, the same region of the PS as well as the FC region were analyzed for all antibodies. To calculate the expression of glycocalyx components, the total length of endothelium surrounding the PS and FC region was measured, and the percentage of GPC4-, SDC4- or HS-positive staining of the total length was calculated for each antibody and region for the individual plaque sections. 

### 4.9. Data Analysis

Statistical analyses were performed using GraphPad Prism 8.3.0. All data sets were tested for normal distribution using the Shapiro–Wilk and Kolmogorov–Smirnov test. Depending on those results, either parametric (passed normality and equivalence variance test) or nonparametric test (failed normality and equivalence variance test) were performed. Protein expression as well as mRNA expression was determined using either flow cytometry or RT-qPCR. Data were log-transformed wherever it was indicated. A random effect model was first fitted to account for within-subject correlation; then, a linear regression model was fitted in cases in which there was no evidence for the random effect. 

The effects on protein expression measured by immunofluorescence were analyzed for two independent variables: (1) shear stress conditions created by two flow types (laminar vs. non-uniform) and (2) exposure time (1 vs. 10 days). The effects on THP-1 cell adhesion were also analyzed for two independent variables: (1) shear stress conditions created by two flow types (laminar vs. non-uniform) and (2) treatment type (control vs. siRNA, heparinase III, or surfen hydrate). Data were analyzed using 2-way ANOVA with the Holm–Sidak post hoc test where the normality and equivalence variance test were passed. Otherwise, data were first divided into two groups according to the flow types and exposure time or treatment types, and the new data sets were compared using either the paired t-test or Wilcoxon signed rank test. The four groups (combination of 2 classes of flow type and 2 classes of exposure time or treatment type) were then compared using 1-way ANOVA with the Holm–Sidak post hoc test (if data passed normality and equivalence variance test) or using the Kruskal–Wallis test with Dunn’s post hoc test (if data failed normality or equivalence variance test). In addition, the effects on immunofluorescence signal intensity were analyzed for two independent variables: (1) shear stress condition (laminar vs. non-uniform) and (2) sex (female vs. male), and analyzed as described above. In this case, the comparison was performed within each exposure time group. Correlations among different glycocalyx markers were analyzed, measuring Pearson’s or Spearman’s correlation coefficient. Differences between the two PS regions were analyzed by the paired *t*-test. Comparing more than two groups (e.g., among different plaque type sections) ordinary 1-way ANOVA/Kruskal–Wallis-test or 2-way ANOVA with appropriate post hoc tests for multiple comparisons were performed. *p* < 0.05 was considered statistically significant in all analyses.

## 5. Conclusions

Our study demonstrated the characteristics of different glycocalyx molecules expressed in ECs in vitro and in vivo. The dynamic adhesion assay showed a potential role of GPC4 in the early stage of atherogenesis by attenuating early adhesion of immune cells to ECs. The distinct expression of GPC4 and HS in human atherosclerotic lesions spotlighted the impact on inflammation and plaque vulnerability. Although further studies are needed to shed light on distinct mechanisms, our data indicated a clinical significance of the endothelial glycocalyx in cardiovascular disease.

## Figures and Tables

**Figure 1 ijms-24-11595-f001:**
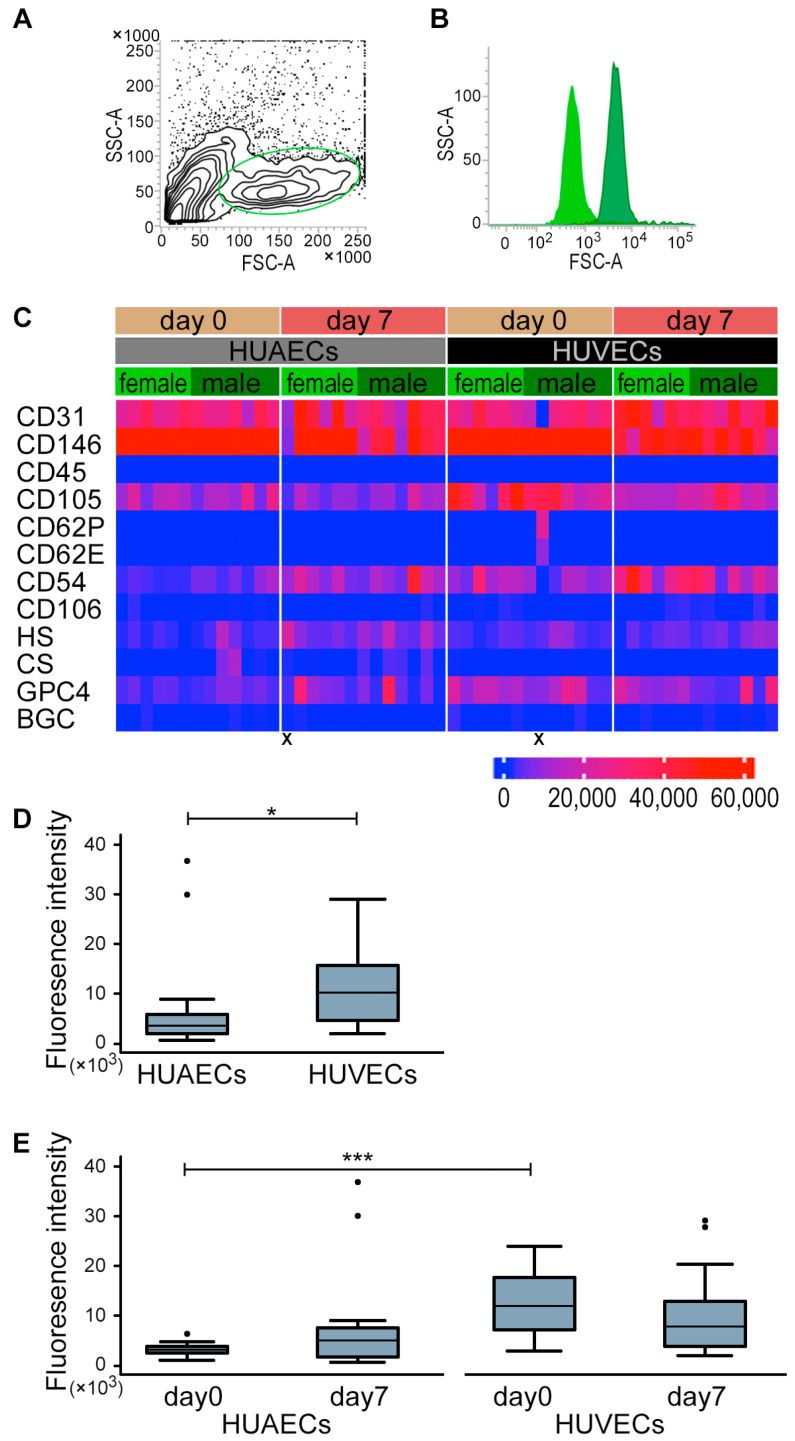
Protein expression in HUVECs and HUAECs. A representative scatterplot of HUVECs (**A**) by flow cytometry. A green circle designates the gate according to SSC and FSC. A representative histogram of GPC4 positive cells detected by flow cytometry (**B**). Light green shows the signal from isotype control, and dark green from anti-GPC4 antibody. Fluorescent intensity of GPC4 in HUAECs and HUVECs (**C**). GPC4 had higher expression in HUVECs vs. HUAECs. Mean fluorescent intensity from 10,000 cells labelled with 12 antibodies, standardized by isotype control, and color-coded. HUAECs and HUVECs were stained with CD31, CD146 (both are positive controls for ECs), CD45 (negative control for HUVECs), CD105 (hematopoietic stem cell marker), CD62P (P-selectin), CD62E (E-selectin), CD54 (ICAM-1), CD106 (VCAM-1), HS (heparan sulfate), CS (chondroitin sulfate), GPC4, and BGC (biglycan). The two samples marked “X” were excluded from further analyses due to unstained positive controls. Expression of GPC4 measured by fluorescent intensity (**D**,**E**). GPC4 expression in HUAECs and HUVECs, regardless of the incubation time or sex (**D**), and the expression between incubation time within HUAECs (**E**, left) and HUVECs (**E**, right). Data are expressed as median with 25th and 75th percentiles, with whiskers indicating 5th and 95th percentiles. Sample size was *n* = 6 (females) and *n* = 7 (males). Significant differences in (**D**,**E)** were evaluated by a mixed linear model with log-transformed data in GPC4. * *p* < 0.05, *** *p* < 0.001. FSC: forward scatter; SSC: side scatter; HUAECs: human umbilical vein endothelial cells; HUVECs: human umbilical artery endothelial cells.

**Figure 2 ijms-24-11595-f002:**
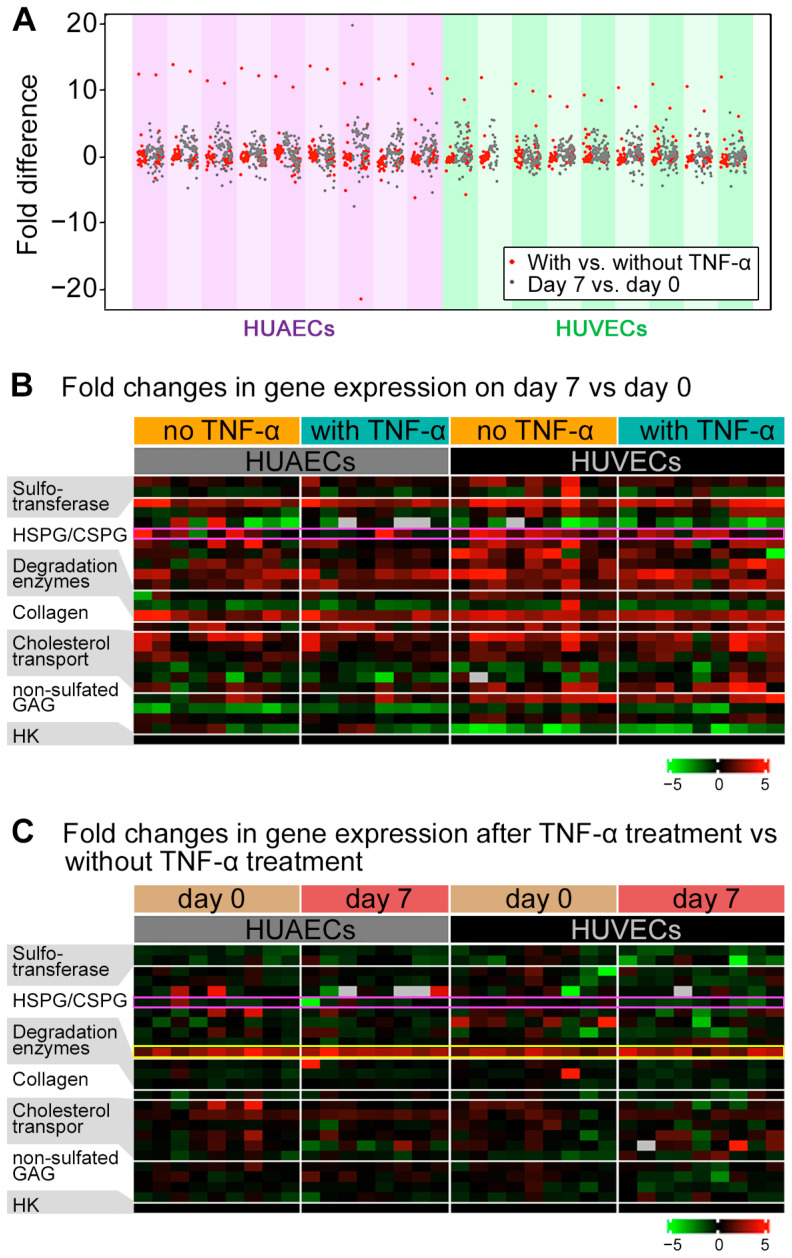
mRNA expression in HUAECs and HUVECs (*n* = 9 each) with and without TNF-α stimulation. Individual data distribution for all 26 genes in HUAECs and HUVECs (**A**). Red dots represent gene expression in TNF-α-stimulated ECs in fold-comparison with those in ECs without TNF-α stimulation. Gray dots represent gene expressions in ECs incubated for 7 days after 80% confluence in fold-comparison with those in ECs stopped for incubation at 80% confluence. Each colored column represents a sample of either HUAECs (purple) or HUVECs (green) from an individual donor. Most of the investigated mRNA showed within five-fold difference between HUAECs and HUVECs or with and without TNF-α. Outliers were excluded from statistical analyses. Fold changes in gene expression of ECs incubated for 7 days, relative to those for 0 days after confluence in HUAECs (*n* = 9) and HUVECs (*n* = 9) (**B**). The fold changes are calculated within cells stimulated with TNF-α (orange) and within cells without TNF-α stimulation (green). GPC4 is framed with a pink line. Fold changes in gene expression of ECs with TNF-α treatment, relative to those without TNF-α treatment in HUAECs (*n* = 9) and HUVECs (*n* = 9) (**C**). The fold changes are calculated within cells incubated 0 days (beige) after confluence and within cells incubated 7 days (pink) after confluence. GPC4 is framed with a pink line, and SDC4 is framed with a yellow line. For both (**B**,**C**), investigated genes are grouped according to the functions (Each row represents (from top to bottom) CHST15 and CHST7 (Sulfotransferase); BGN, GPC1, GPC3, GPC4, HSPG2, MXRA5, SDC1, SDC2, and SDC4 (HSPG/CSPG); GALNS, HPSE1, and MMP2 (degradation enzymes); COL4A5 (collagen); ABCA1, ABCG1, LCAT, LDL-R, PCSK9-2, and VLDL-R (cholesterol transport); CD44, HYAL1, HYAL2, and LYVE1 (non-sulfated GAG); and YWHAZ (housekeeping gene)). HUAECs: human umbilical vein endothelial cells; HUVECs: human umbilical artery endothelial cells; TNF-α: tumor necrosis factor-α; HSPG: heparan sulfate proteoglycan; CSPG: chondroitin sulfate proteoglycan; HK: housekeeping.

**Figure 3 ijms-24-11595-f003:**
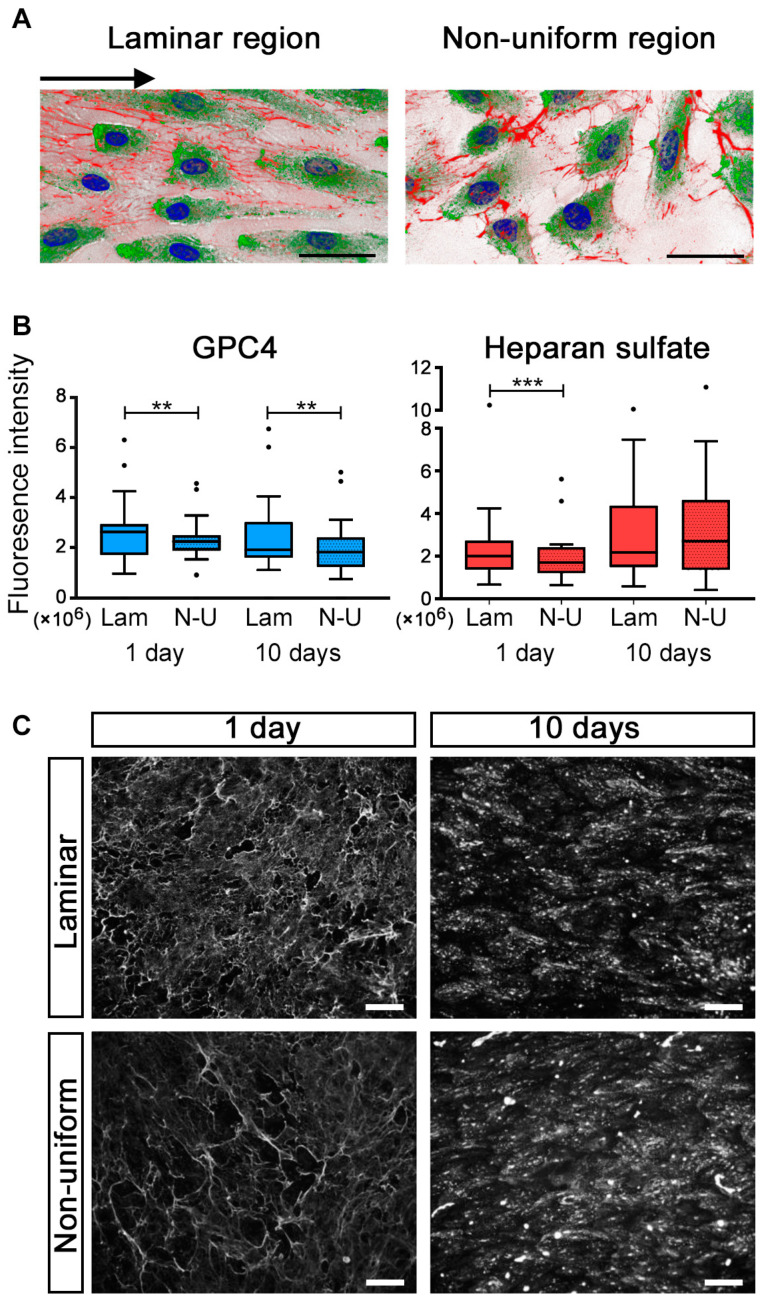
Expression of GPC4 in HUVECs under flow conditions. Representative microphotographs of HUVECs (**A**) at laminar and non-uniform regions, immunostained for GPC4 (green), HS (red) and nuclei (blue). The arrows designate flow direction. Scale bars, 40 µm. Fluorescent intensity was quantified for GPC4 and HS in cells exposed to either a 1-day (*n* = 24) or 10-day flow (*n* = 25) (**B**). Data are expressed as median with interquartile range (IQR: first and third quartiles), with whiskers indicating 1.5 × IQR. GPC4 had a trend to express more highly at laminar compared with non-uniform regions at both time points, while this was only seen after 1 day of flow in HS expression. Differences between laminar and non-uniform regions were compared using the paired *t*-test (GPC4) or Wilcoxon signed rank test (HS), and those between 1-day vs. 10-day flow exposure by the Kruskal–Wallis test). ** *p* < 0.01, *** *p* < 0.001. Lam = laminar; N-U = non-uniform. Different flow- and time-dependent HS expression pattern in HUVECs (**C**). Scale bars, 100 µm.

**Figure 4 ijms-24-11595-f004:**
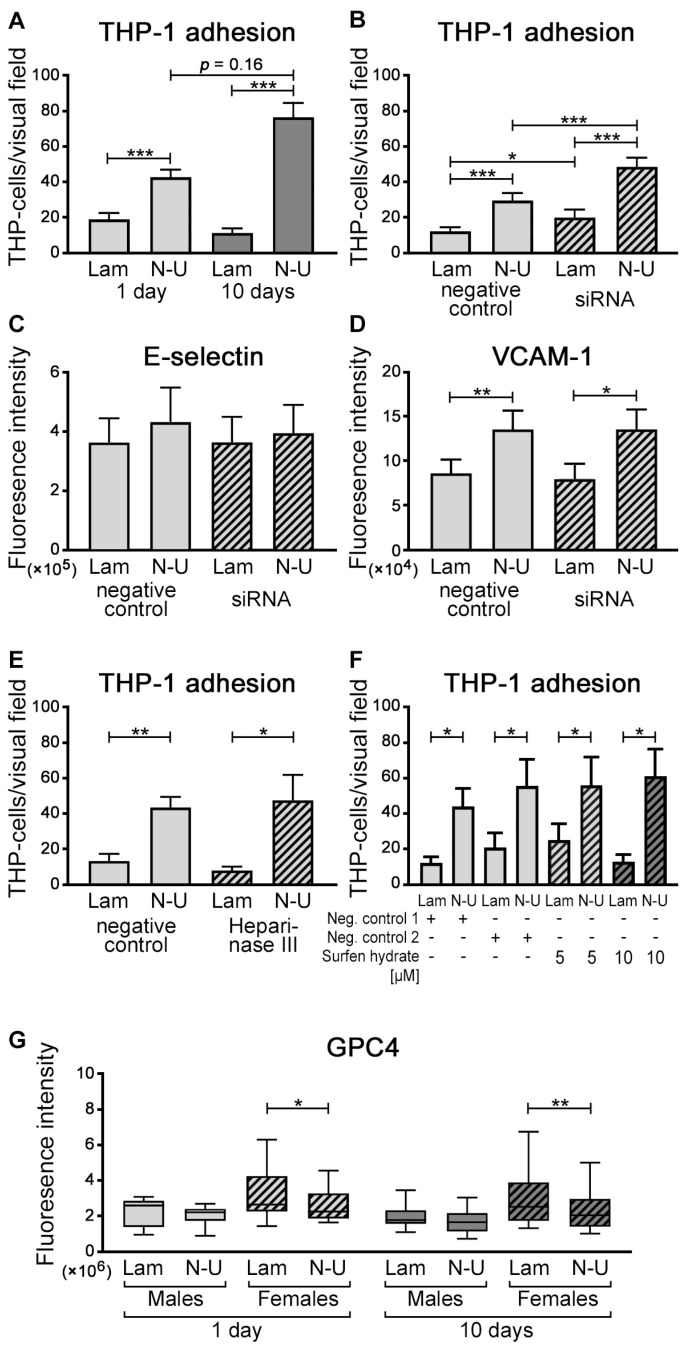
THP-1 adhesion to endothelial cells. Using dynamic adhesion assay, adhesion of THP-1 cells to TNF-α–stimulated HUVECs was measured. Adhesion was compared between HUVECs exposed to 1-day (*n* = 29) or 10-day (*n* = 28) flow (**A**). Adhesion was significantly higher at non-uniform compared with laminar flow regions, which was more eminent in HUVECs exposed to 10-day flow (paired *t*-test/Wilcoxon signed rank test). Comparison between 1-day and 10-day flow exposure revealed higher adhesion after 10 days in non-uniform regions (Kruskal–Wallis test; *p* = 0.16). When GPC4 was knocked down in HUVECs (**B**), the adhesion at non-uniform regions was significantly higher compared with the negative control (*n* = 13; two-way ANOVA with repeated measures with Sidak post hoc test; *p* < 0.001), while the knockdown had no significant effect on adhesion molecule expression of E-selectin (**C**) and VCAM-1 (**D**) (*n* = 6 for each; paired *t*-test between laminar and non-uniform regions and Kruskal–Wallis test between negative control and siRNA). Heparinase III treatment (*n* = 5) (**E**) or competitive HS antagonist surfen hydrate (*n* = 6) (**F**) had no influence on THP-1 cell adhesion (paired *t*-test between laminar and non-uniform regions and Kruskal–Wallis test between negative control and heparinase III treatment). GPC4 tends to be more highly expressed in female samples at both laminar and non-uniform regions the after 1-day flow (*n* = 11 for female and *n* = 13 for male), but not after the 10-day flow exposure (*n* = 11 for female and *n* = 12 for male) (paired *t*-test between laminar and non-uniform regions and Kruskal–Wallis test between male and female) (**G**). * *p* < 0.05, ** *p* < 0.01, *** *p* < 0.001. Lam: laminar; N-U: non-uniform; VCAM-1: vascular cell adhesion molecule-1; GPC4: glypican-4. Data are expressed as mean ± SEM (**A**–**C**) or as median with interquartile range (IQR), with whiskers indicating 1.5 × IQR (**D**).

**Figure 5 ijms-24-11595-f005:**
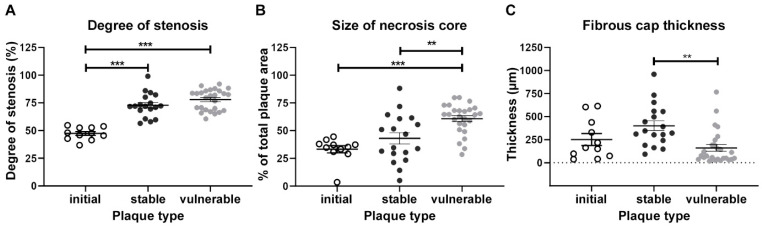
Comparison of degree of stenosis (**A**), size of necrotic core (**B**). and the FC thickness (**C**) among different plaque type sections. The area of the whole plaque and the necrotic core as well as the FC thickness at the thinnest site were measured using NIS Elements^®^ software version 5.21.03 from trichrome stained plaques (*n* = 55), and the percentage of the necrotic core and the degree of stenosis were calculated. Data are shown as mean ± SEM with individual values (**A**,**B**), or individual values with median (**C**). Comparison was done using one-way ANOVA (A) or the Kruskal–Wallis test (**B**,**C**) with appropriate post hoc tests. v: vulnerable; s: stable; i: initial. ** *p* < 0.01, *** *p* < 0.001.

**Figure 6 ijms-24-11595-f006:**
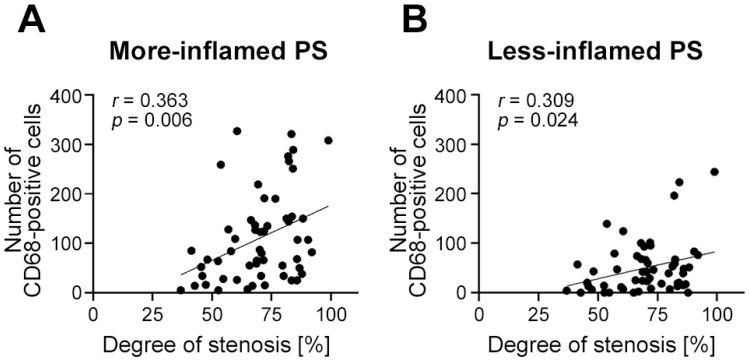
Correlations between the degree of stenosis and the number of CD68-positive cells. Pearson correlation analyses were performed in all plaque sections (*n* = 55) for the more-inflamed (**A**) and less-inflamed (**B**) PS. Graphs show simple linear regression. PS: plaque shoulder; *r*: Pearson coefficient.

**Figure 7 ijms-24-11595-f007:**
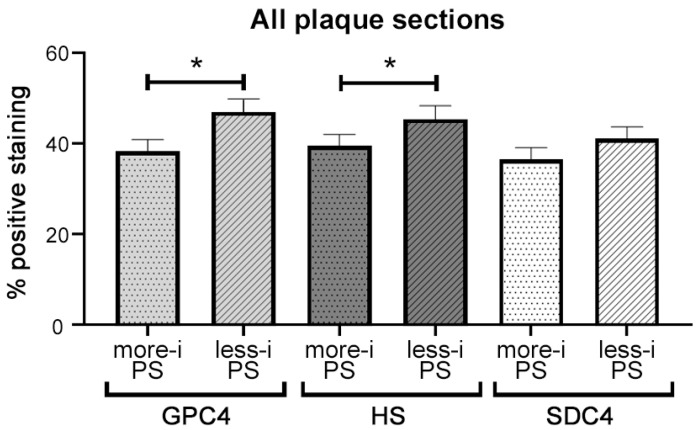
Expression of GPC4, HS, and SDC4 in all plaque sections. Images of both PSs were taken at 150× magnification with an Olympus IX70 inverted microscope and analyzed with NIS elements^®^ software version 5.21.03. The total length of the PS and the positively stained endothelium were measured, and the percentage of positively stained endothelium was calculated for every picture. Data are shown as mean ± SEM (*n* = 55). Comparison between the two PSs was done using paired *t*-test. * *p* < 0.05. GPC4: glypican-4; HS: heparan sulfate; SDC4: syndecan-4; more-i PS: more-inflamed plaque shoulder; less-i PS: less-inflamed plaque shoulder.

**Figure 8 ijms-24-11595-f008:**
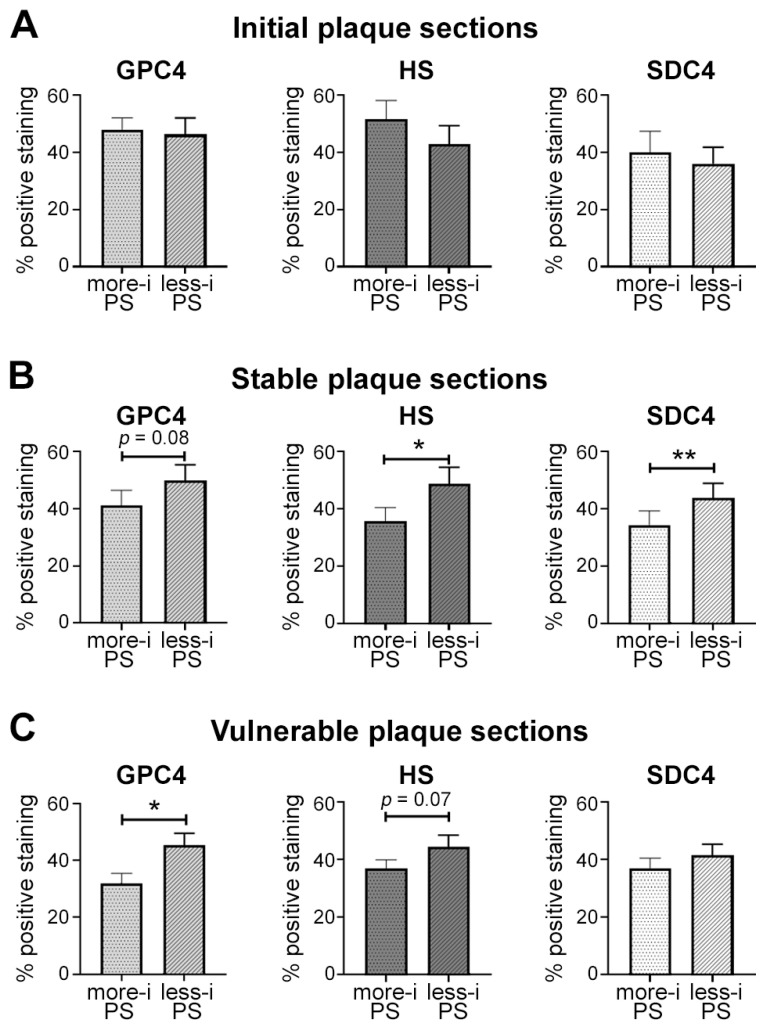
Expression of GPC4, HS, and SDC4 in initial (**A**), stable (**B**), and vulnerable (**C**) plaque sections. Percentage endothelial expression of the analyzed glycocalyx components was assessed as described. Data are shown as mean ± SEM. Paired *t*-test (GPC4, SDC4, HS (vulnerable, stable), or Wilcoxon matched pairs signed rank test (HS (initial)) was used for PS comparison. *n* = 26 (vulnerable plaque sections); *n* = 18 (stable plaque sections); *n* = 11 (initial plaque sections). * *p* < 0.05, ** *p* < 0.01. GPC4: glypican-4; HS: heparan sulfate; SDC4: syndecan-4; PS: plaque shoulder; more-i PS: more-inflamed plaque shoulder; less-i PS: less-inflamed plaque shoulder.

**Figure 9 ijms-24-11595-f009:**
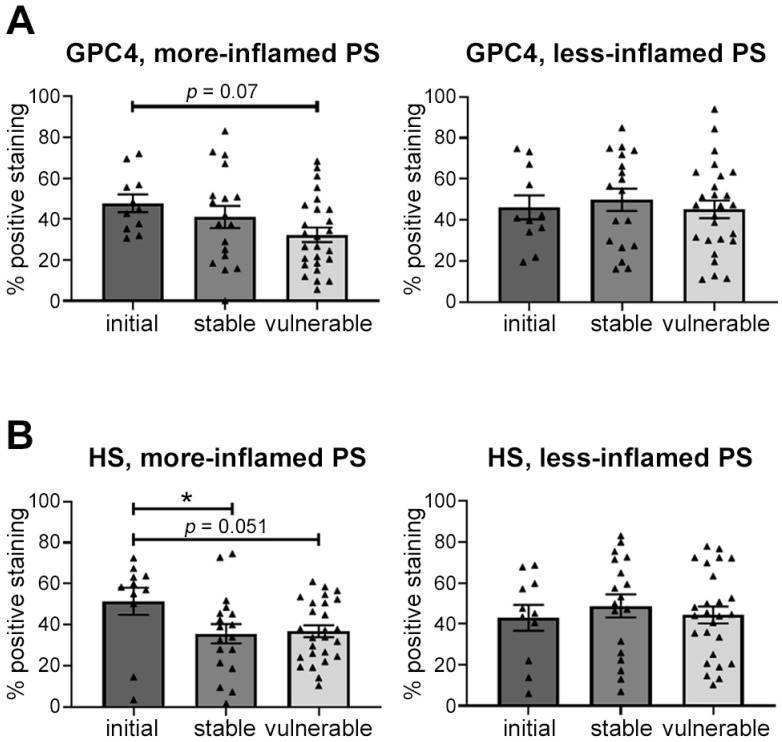
Comparison of the expression of GPC4 (**A**) and HS (**B**) among different plaque-type sections. Percentage endothelial expression of the analyzed glycocalyx components was assessed as described. Data are shown as mean ± SEM with individuals. Comparison among initial (*n* = 11), stable (*n* = 18), and vulnerable (*n* = 26) plaque sections was done using one-way ANOVA with appropriate post hoc test. * *p* < 0.05. GPC4: glypican-4; HS: heparan sulfate; PS: plaque shoulder.

**Figure 10 ijms-24-11595-f010:**
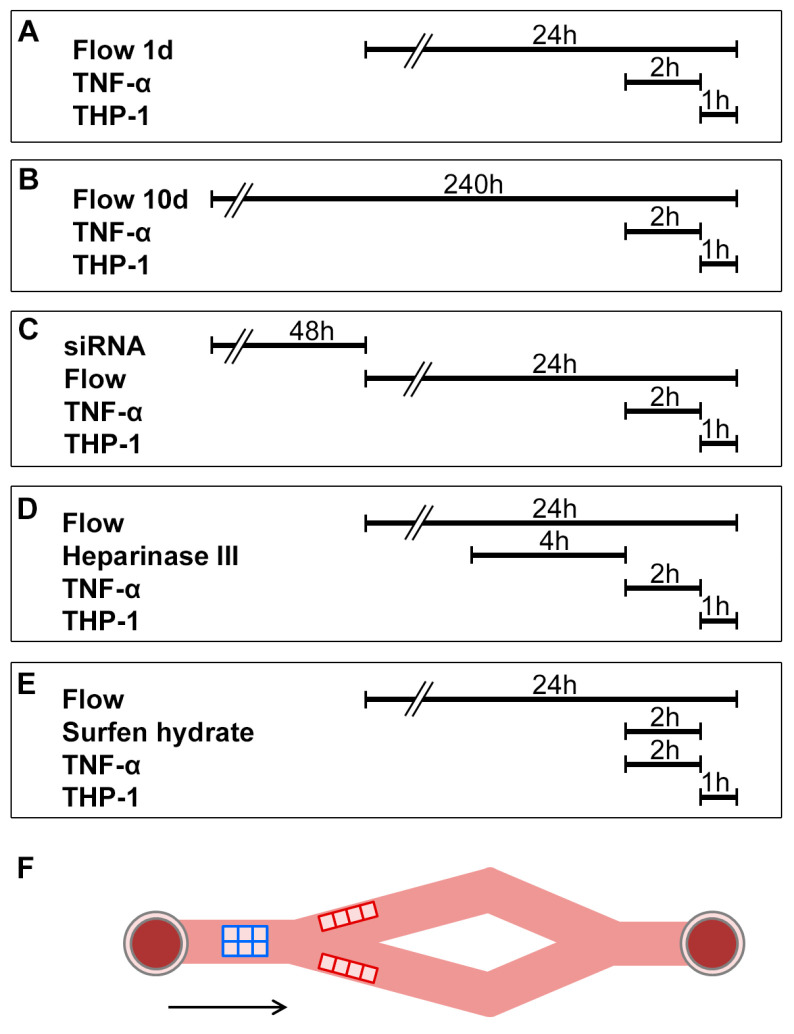
Flow experiment design (**A**–**E**) and scheme of a flow-chamber slide (**F**). In the dynamic adhesion assay, flow (9.6 mL/min) was applied for 24 h for 1-day experiments (**A**) and for 240 h for 10-day experiments (**B**), with a TNF-α stimulation and THP-1 adhesion at the end of the process. To silence GPC4, siRNA against GPC4 was transfected 48 h before the flow experiment (**C**). Heparinase III was added 6 h before THP-1 addition to degrade HS (**D**), and surfen hydrate was added simultaneously with TNF-α for 2 h followed by adhesion of THP-1 cells (**E**). Schematic of flow-chamber slide showing the locations analyzed (**F**). Micrographs were taken at 6 visual fields at laminar regions (blue boxes) and 8 visual fields at non-uniform regions (red boxes). The arrow designates the direction of the flow. TNF-α: tumor necrosis factor-α.

**Table 1 ijms-24-11595-t001:** Patient characteristics.

	All (%)	Male (%)	Female (%)	Odds Ratio [95% CI]	*p*-Value
Patients	55 (100)	38 (69)	17 (31)	-	
Age (median ± SD)	75 ± 8.9	75 ± 9.2	70 ± 8.2	-	0.385 ^a^
BMI (mean ± SEM)	27.7 ± 0.7	28.3 ± 0.8	26.4 ± 1.4	-	0.099 ^a^
Obesity	16 (29.1)	11 (28.9)	5 (29.4)	1.02 [0.3126–3.259]	1.0 ^b^
Smoker *	25 (45.5)	17 (44.7)	8 (47.1)	1.10 [0.3799–3.303]	0.89 ^c^
T2D	19 (34.5)	15 (39.5)	4 (23.5)	0.47 [0.5561–1.798]	0.36 ^b^
Hypertension	52 (94.5)	36 (94.7)	16 (94.1)	0.89 [0.0981–13.61]	1.0 ^b^
Increased lipid serum levels **	47 (85.5)	32 (84.2)	15 (88.2)	1.41 [0.2876–7.450]	1.0 ^b^
Asymptomatic	37 (67.3)	26 (68.4)	11 (64.8)	0.85 [0.2739–2.911]	0.97 ^c^
Symptomatic	18 (32.7)	12 (31.6)	6 (35.2)		

* Former and to date; ** total cholesterol, triglycerides, and low-density lipoprotein. ^a^
*t*-test, ^b^ Fisher’s exact test, ^c^ Chi-square test with Yates’ correction. BMI, body mass index; CI, confidence interval; SD standard deviation; SEM, standard error of mean; T2D, type 2 diabetes.

## Data Availability

The data presented in this study are available on request from the corresponding author.

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
