# Peer review of "The Shear Stress–Regulated Expression of Glypican-4 in Endothelial Dysfunction In Vitro and Its Clinical Significance in Atherosclerosis"

_ijms, 2023, doi:10.3390/ijms241411595_

Round 1

Reviewer 1 Report

The manuscript lacks recent references. The data attached are not very promising. GPC4 and HS are involved in atherosclerosis, and their distinctive mechanism should be explored and discussed. The manuscript has certain grammatical errors.

Reviewer 2 Report

The authors analyzed the expression of the glycocalyx component, glypican 4, by venous and arterial endothelial cells in vitro, including flow conditions. Using a model system as well as carotid plaque specimens taken during carotid endarterectomy, the original data confirming the atheroprotective role of these molecules were obtained. The results suggest the involvement of GPC4 both in early and advanced stages of atherogenesis.

Comments.

1.  The authors analyzed the expression by HUVEC and HUAEC of a number of genes (by mRNA content). However, I could not find a list of primers, description of PCR conditions, etc.

2.           The source of mRNA against GPC-4 is unclear.

3.           In section 2.1, the authors write "The correlations between the HUVECs marker CD31 and these activation markers were statistically significant..." For what purpose were these correlations tested?

4.           Figure 2 (A) is not clear to me. What does it demonstrate? Individual distribution of which mRNA is given?  

5.           Section 4.4. is titled "Immunocytochemistry" although immunofluorescence is described.

6.           In the Methods section, the authors write "To define the more-inflamed PS, CD68 positive macrophages were counted in a 0.25 mm2 region of interest." What was the criterion for separating more-inflamed/less-inflamed?

7.           In the supplementary material, there is an unmarked figure before Table S4. It should be removed.

Reviewer 3 Report

In this manuscript, the authors studied the expression of glypican-4 by HUVEC and HUAEC under different flow conditions and with TNF-a stimulation. They identified GPC-4 expression is effected by shear stress but not by TNF-a and the expression of level GPC-4 is inversely proportional to the amount of THP-1 adhesion in vitro. In patient samples, GPC4 expression is lower in more inflamed plaque shoulders.

Specific comments:

·         Could the authors comments on why HUAEC was not included in the subsequent experiments?

·         Scale bar is missing for Fig 3A

·         Scale bar length is missing for Fig 3C

·         Could author specify the parameters used for determining initial, stable and vulnerable plaque types?

·         How was more or less inflamed PS determined?

·         Line 251-53, which figure supports this statement?

·         Line 307-308, the r and p values in texts are different than what are on Fig S3.

·         Supplementary - page 5, graph inserted at top of the page by mistake?

Typos and minor grammar mistakes noticed.  
